# The Lack of Systemic and Subclinical Side Effects of Botulinum Neurotoxin Type-A in Patients Affected by Post-Stroke Spasticity: A Longitudinal Cohort Study

**DOI:** 10.3390/toxins14080564

**Published:** 2022-08-19

**Authors:** Marco Battaglia, Margherita Beatrice Borg, Lara Torgano, Alberto Loro, Lucia Cosenza, Michele Bertoni, Alessandro Picelli, Andrea Santamato, Marco Invernizzi, Francesca Uberti, Claudio Molinari, Stefano Carda, Alessio Baricich

**Affiliations:** 1Physical and Rehabilitation Medicine, Department of Health Sciences, Università del Piemonte Orientale, 28100 Novara, Italy; 2Physical and Rehabilitation Medicine, “Ospedale Maggiore della Carità” University Hospital, 28100 Novara, Italy; 3Rehabilitation Unit, Department of Rehabilitation, “Santi Antonio e Biagio e Cesare Arrigo” National Hospital, 15121 Alessandria, Italy; 4Physical Medicine and Rehabilitation, ASST Sette Laghi, 21100 Varese, Italy; 5Department of Neurosciences, Biomedicine and Movement Sciences, University of Verona, 37134 Verona, Italy; 6Neurorehabilitation Unit, Department of Neurosciences, University Hospital of Verona, 37126 Verona, Italy; 7Physical Medicine and Rehabilitation, Spasticity and Movement Disorder Unit, Policlinico Riuniti, University of Foggia, Viale Pinto 1, 71122 Foggia, Italy; 8Dipartimento Attività Integrate Ricerca e Innovazione (DAIRI), Translational Medicine, “Santi Antonio e Biagio e Cesare Arrigo” National Hospital, 15121 Alessandria, Italy; 9Human Physiology, Department of Translational Medicine, Università del Piemonte Orientale, 28100 Novara, Italy; 10Human Physiology, Department of Sustainable Development and Ecological Transition, Università del Piemonte Orientale, 13100 Vercelli, Italy; 11Neuropsychology and Neurorehabilitation Service, Department of Clinical Neuroscience, Lausanne University Hospital, 1004 Lausanne, Switzerland

**Keywords:** muscle spasticity, botulinum toxin, heart rate, stroke, autonomic nervous system, rehabilitation

## Abstract

Botulinum Neurotoxin type-A (BoNT-A) is the treatment of choice for focal post-stroke spasticity (PSS). Due to its mechanism of action and the administration method, some authors raised concern about its possible systemic diffusion leading to contralateral muscle weakness and autonomic nervous system (ANS) alterations. Stroke itself is a cause of motor disability and ANS impairment; therefore, it is mandatory to prevent any source of additional loss of strength and adjunctive ANS disturbance. We enrolled 15 hemiparetic stroke survivors affected by PSS already addressed to BoNT-A treatment. Contralateral handgrip strength and ANS parameters, such as heart rate variability, impedance cardiography values, and respiratory sinus arrythmia, were measured 24 h before (T0) and 10 days after (T1) the ultrasound (US)-guided BoNT-A injection. At T1, neither strength loss nor modification of the basal ANS patterns were found. These findings support recent literature about the safety profile of BoNT-A, endorsing the importance of the US guide for a precise targeting and the sparing of “critical” structures as vessels and nerves.

## 1. Introduction

Li and colleagues recently re-defined spasticity as “velocity- and muscle length-dependent increase in resistance to externally imposed muscle stretch. It results from hyperexcitable descending excitatory brainstem pathways and from the resultant exaggerated stretch reflex responses” [1]. Spasticity is among the most severe clinical expressions of the upper motor neuron syndrome (UMNS), and stroke represents one of the major causes of UMNS, with a global incidence of approximately 258/10,000/year [2]. In Europe, 1.1 million people suffer from a stroke each year [3], and it represents the fifth-leading death cause and the leading long-term disability cause in the US [4].

PSS more frequently affects the upper limb than the lower limb, with an incidence that varies from 7 to 38% in the upper limb during the first 12 months after the acute event [5,6].

Alongside with PSS, several signs and symptoms characterize stroke survivors in the subacute and chronic phase. Among these, pain, fatigue, paresis, sensorimotor impairment, depression, and cognitive dysfunction mostly contribute to an impaired quality of life (QoL) [7,8].

Regarding fatigue, it was described by Staub et al. as “a feeling of early exhaustion with weariness, lack of energy and aversion to effort that develops during physical or mental activity and is usually not ameliorated by rest” [9]. The prevalence of fatigue in stroke survivors ranges between 30 and 77%, and, alongside depression [10], it negatively affects both physical and psychological rehabilitation programs, leading to a significantly increased chance of relapsing neurovascular events. Furthermore, increasing the risk of cardiovascular complications, it is an important death predictor after stroke onset [11,12]. To this end, it is crucial to finely tailor the rehabilitation program to prevent and contain fatigue expression.

Notably, PSS affects all the three domains of the International Classification of Functioning, Disability and Health (ICF) of the World Health Organization (WHO) [13], resulting in reduced overall function; hence, lower levels of independence. Therefore, PSS treatment is a fundamental element in the multimodal management of hemiparetic stroke survivors.

The clinical management of PSS relies on an accurate physical examination and assessment of the spasticity grade, measured with the Modified Ashworth Scale (MAS) [14,15] or the Tardieu scale [16]. The treatment of choice consists in the intramuscular injection of Botulinum Neurotoxin type-A (BoNT-A), in close association with several non-pharmacological and/or surgical interventions [17,18,19,20,21,22].

BoNT-A already proved to be an effective and safe treatment for focal spasticity, providing a transient chemodenervation of injected muscles throughout the inhibition of the pre-synaptic release of acetylcholine at the neuromuscular junctions [23]. Together with its high efficacy and optimum safety profile [24], BoNT-A is specifically proved to be effective in reducing pain, improving mobility, walking ability, activities of daily living, and quality of life [25,26]. Therefore, BoNT-A is currently considered the gold standard treatment in patients affected by focal PSS.

Needless to say, it is crucial to assess the possibility of BoNT-A-induced local and general adverse reactions which have been reported in the scientific literature [27,28]. Systemic BoNT-A spread can be due to a combination of vascular and neural diffusion, leading to potential clinically relevant conditions. BoNT-A diffusion, spread, and migration [29] might lead to contralateral muscle weakness, dysphagia, dysarthria, and several autonomic nervous system (ANS) subclinical impairments, such as dry mouth; postural hypotension with consequent falls; constipation; and, particularly, cardiorespiratory drive alterations [30,31]. These incidental events, in fact, may seriously worsen already severely compromised patients affected by underlying weakness, fatigue, and stroke-induced ANS imbalance [32].

Commonly, validated tools are used to evaluate these potential alterations. For segmentary strength, the most feasible technique is the handgrip strength test [33]. Regarding ANS, a multiparametric approach is preferred, taking into consideration the heart rate variability (HRV) oscillations [34,35], the impedance cardiography (ICG) panel [36], and the respiratory sinus arrythmia (RSA) variations [37]. These values can be obtained through specific electrocardiographic (ECG) registrations.

Currently, these side effects have been only investigated on small populations, mainly in relation to high-dose administration, giving nonetheless promising results and so far confirming the high safety profile of BoNT-A [38,39]. Throughout the inclusion of subjects treated with lower doses of BoNT-A and considering the three main formulations (Incobotulinumtoxin A, Abobotulinumtoxin A, and Onabotulinumtoxin A), this study aims to investigate an additional population with a larger sample size.

The aim of this study is to assess whether BoNT-A treatment is associated with systemic side effects, such as contralateral weakness, or subclinical ANS alterations, as a consequence of systemic diffusion of the locally intramuscular-injected BoNT-A in hemiplegic stroke survivors affected by PSS.

## 2. Results

Seventy-five patients were pre-screened in order to participate in the study; 23 patients respected the inclusion and exclusion criteria. After losing eight patients at the follow-up, 15 subjects completed the study. All the patients were treated with BoNT-A, using one of the three commercialized drugs in Italy, as shown in Table 1.

The dose chosen for every injection site for each patient is reported in Table 2. In nine patients, muscles in both the upper and the lower limb were treated; in four patients, only in the upper limb; and in two, only in the lower limb.

The results of the handgrip strength test and the electrocardiographic registration of HRV, ICG, and RSA parameters showed no statistical differences between T0 and T1 after BoNT-A injections, as fully described in Table 3.

Subgroup analysis was implemented in order to compare higher-dose (700–800 UI of OnabotulinumtoxinA or 1500 UI of AbobotulinumtoxinA) with lower-dose subjects (other doses). No significant difference of parameter variations from T0 to T1 was found between the two groups, as seen in Table 4.

## 3. Discussion

Our results show no significative differences in every investigated variable before and after BoNT-A injections. Specifically, no variations of contralateral handgrip strength were found. Similarly, HRV, pre-ejection period (PEP), dz/dt min position, dz/dt min value, and respiratory sinus arrhythmia (RSA) parameters showed no discrepancies after BoNT-A treatment.

Additionally, the subgroup analysis confirmed the absence of a significant alteration of the variables from T0 to T1 between high- and low-dose patients, endorsing the hypothesis of the non-significant systemic side effects of BoNT-A, even with maximum dosages.

However, the nature of BoNT-A as a bacterial toxin and its intramuscular way of administration through multiple injections raised concerns about its possible systemic diffusion and adverse effects. An internal spread has already been described inside the injected muscle, together with its diffusivity in adjacent muscular masses, trespassing aponeurosis, and also to the contralateral muscles [40,41]. However, to date, neither the entity nor the mechanism of botulinum toxin diffusion outside the injection site are well known. Various pathways have been hypothesized, including systemic spread, vascular diffusion, and retrograde axonal spread [30,41].

The potential systemic and subclinical adverse effects of BoNT-A treatment represent a crucial issue that requires deep investigations given the increasing trend of BoNT-A employment, thus representing a hot topic in rehabilitation.

In the scientific literature, there is not a univocal point of view. In more detail, in 2012, Thomas and colleagues described in two stroke survivors affected by spastic hemiparesis, previously inoculated with BoNT-A at the upper limb, the onset of contralateral weakness. After electromyographic examination, the presence of a neuro-muscular blockade in the non-treated contralateral limb muscles was revealed. Successively, weakness spontaneously recovered in few months [42]. Another more recent report published in 2020 concerned a patient who, after BoNT-A injections, showed paresis affecting the contralateral deltoid muscle. Even in this case, the neuromuscular blockade was proved through an electromyographic investigation, and the recovery was complete after 8 months. In this case, it is quite interesting that the contralateral paresis developed after the therapeutic switch from incobotulinum to abobutolinum toxin. Thus, the authors suggested extreme caution when converting from one molecule to another [30]. Luckily, these adverse events are uncommon and, indeed, in our larger study sample, we did not detect a variation of contralateral upper limb strength after BoNT-A effect onset.

Given the highly complex disability condition of stroke survivors needing a focal spasticity treatment, it is crucial to prevent any possible strength loss in the healthy limbs. Moreover, taking into consideration the frequency of post-stroke fatigue and its poor prognostic meaning [9,10,11,12], it is mandatory to avoid any source of possible treatment-induced weakness that could further severely worsen an already debilitated subject. Therefore, it is of primary importance to determine whether BoNT-A treatment could provoke contralateral weakness and/or paresis in bigger study samples, considering that this side effect may negatively impact the rehabilitation programs, the activities of daily living, and, consequently, the QoL of these patients. To this end, our results are promising, supporting BoNT-A safety.

Considering the ANS evaluation, HRV already represents a reliable marker which predicts the recurrence of stroke and cardiovascular accidents, post-stroke complications, and functional outcomes [34]. HRV is a physiological phenomenon finely regulated by several agents: at first, the sympathetic and parasympathetic balance; then, the respiration rate and depth, the baroreceptors reflex, and the rhythmic changes in vascular tone [35]. The combination of these mechanisms generates a constant oscillation typical of a healthy organism. Recent literature discusses how the neurological lesions after stroke can impoverish these physiological oscillations, and represent a major cause of ANS dysfunction, frequently leading to cardiovascular drive alterations [32]. Therefore, the role of BoNT-A in potentially negatively affecting this already altered condition needs to be clarified.

In fact, BoNT-A, due to its mechanism of action, may affect the parasympathetic nervous system and, thus, it also may impact cardiac regulation. The inhibition of the pre-synaptic release of acetylcholine could prevent the activation of M2 muscarinic receptors localized in the cardiac muscle, which are in charge of the vagal inhibition transmission. As reported by Girlanda et al., BoNT-A could then be responsible for subclinical changes in heart rate [43], which can be detected through HRV analysis.

HRV is described by time domains that quantify the variability of the interbeat intervals (IBIs), and frequency domains. Time domains include: IBIs number, IBIs average, standard deviation of normal-to-normal intervals (SDNN), heart rate standard deviation (HR SD), and root mean square of successive R-R intervals differences (RMSSD) [35]. Frequency domains, though, are calculated by a spectrum analysis based on the Fourier transform, and are used to estimate the amount of absolute or relative signal energy found in different frequency bands. Spectrum components are classified in ultra-low-frequency (ULF), very-low-frequency (VLF), low-frequency (LF), and high-frequency (HF) bands. In our paper, we investigated only LF and HF because low-frequency (0.04–0.15 Hz) bundles mainly depend on the sympathetic nervous system and baroreceptor action, whereas the high-frequency (0.15–0.4 Hz) component is mostly under vagal control, indicating parasympathetic activity. Therefore, the ratio LF/HF expresses the balance between the sympathetic and parasympathetic nervous system on heart rate frequency variations [35].

Our results show no alteration of the autonomic drive after BoNT-A injection, in both time and frequency domains. Reviewing the current scientific literature, concordant findings are reported by Baricich et al. and Invernizzi et al. regarding the impact of high doses of IncobotulinumtoxinA and OnabotulinumtoxinA. In both cases, no differences of HRV subsequent to BoNT-A injections were found, even with total doses over 600 IU [38,39], supporting the thesis that focal spasticity treatment with botulinum toxin does not seem to affect the ANS activity.

In addition to the ECG traces, VU-AMS equipment allows the recording of ICG waves, obtained and analyzed through the software, VU-DAMS version 4.0. Specifically, the ICG parameters considered are the pre-ejection period (PEP), the dZ/dt min position, and the dZ/dt min value [36]. PEP reflects the left ventricular contractility representing the duration of the electric or isovolumic systole, and it is commonly used as a marker of sympathetic nervous system activation. The dZ/dt min position and volume parameters, representing, respectively, the minimum value of cardiac and thoracic impedance, give an indirect measure of left ventricle stroke volume.

These parameters measure the ANS influence on the cardiac muscle, and they are, therefore, a feasible tool to assess possible variations of contractility after BoNT-A inoculation.

Even in this case, no post-injection alteration was recorded in our population.

Finally, RSA is a further indirect index of vagal activity on heart rate. As inspiration is responsible for the cyclic suppression of the vagal tone, during this respiratory phase, it is physiological to observe a cardiac frequency increase [37]. Therefore, RSA and RSA-0 (which is the point where RSA value is 0) identify the balance between cardiac and respiratory function. RSA plays a major role in timing alveolar ventilation and perfusion. Reducing the heart rate during expiration is a form of energy-saving procedure, since it suppresses unnecessary heartbeats and, consequently, pulmonary blood flow, whereas the alveoli are in a low air volume condition.

Taking into consideration our study sample, characterized by a chronic disability and energy consumption imbalance, it is a primary issue to maintain the natural variability of the cardiopulmonary system under the influence of ANS as physiologically as possible. Even in this case, our results support the preservation of the physiological fluctuation of RSA after BoNT-A injection.

In conclusion, we suggest that there are no considerable influences of the locally injected botulinum toxin on the ANS activity. Therefore, we can assume that the treatment of focal post-stroke spasticity with BoNT-A does not seem to imply notable subclinical signs or symptoms of systemic diffusion in our population, even considering high doses that are frequently required to reach an adequate therapeutic effect.

The authors finally suggest the use of the US guide to perform BoNT-A injections. Given the potential vascular and axonal spread and the possible trans-aponeurosis diffusion [29], it should be common practice to perform the injection under a real-time US guide in order to obtain a precise targeting [44] and keep a safe distance from vascular, neural, and fascial structures.

The authors are aware of the limitations of this study.

Firstly, the study sample is quite small, even though the current literature is based on even smaller groups. Regardless, it could be difficult to generalize our results to the entire stroke survivor population.

Secondly, the panel of ANS parameters we considered is wide, but not complete. It is advisable to take into account other variables, such as the continuous recording of blood pressure and peripheral oxygen saturation. Regardless, the current literature supports the prognostic and predictive value of the values we considered.

## 4. Conclusions

Our results support the updated scientific literature about the BoNT-A safety profile in stroke survivors affected by spastic hemiparesis.

In our study sample, we did not observe any subclinical sign of systemic BoNT-A effects. In fact, none of our patients showed significant variations of contralateral upper limb strength and of the considered autonomic nervous system parameters from basal, even in case of high-dose treatment. Therefore, in the context of post-stroke focal spasticity treatment, we can endorse the hypothesis that BoNT-A US-guided intramuscular injections do not seem to imply systemic subclinical autonomic or muscular side effects.

Further investigations are needed in order to better clarify these findings in the general post-stroke population affected by spastic hemiparesis undergoing BoNT-A treatment. The extension of the follow-up to one month after injection may provide adjunctive data, considering the profile of the BoNT-A pharmacological effects. Future research may take into consideration a wider board of ANS parameters, hopefully with more and more accurate detection means, and multidistrict muscle strength evaluations.

## 5. Materials and Methods

This is a single-centered, longitudinal, observational study set in the Physical Medicine and Rehabilitation Unit of the University Hospital Maggiore della Carità-Novara (28100 Novara, Italy). We enrolled the study population among chronic stroke survivors affected by spastic hemiparesis already addressed as outpatients to our Center for the periodical clinical re-evaluation and eventual treatment with BoNT-A. No naïve patients were included in order to portray the effect of a chronic treatment.

Study participation was subordinate to the clinical indication to perform the treatment of focal spasticity with BoNT-A. The inclusion criteria were age ≥ 18 years, ischemic or hemorrhagic stroke (clinically and radiologically documented), and focal spasticity with MAS ≥ 2 affecting the upper or the lower limb. Exclusion criteria were former cardiac failure (class ≥ 3 of the New York Heart Association, NYHA [45]), previously diagnosed cardiac arrythmia, ongoing treatment with beta-blockers, the presence of a pacemaker, and permanent contractures affecting the BoNT-A target muscles.

All the participants gave their written informed consent for study participation, structured according to the Declaration of Helsinki, and validated by the local Ethics Committee (CE registration number 160/21, Eudract number: 2019-001834-33) and the Competent Authority (Maggiore della Carità University Hospital, Novara, Italy Protocol 0016937/21, validated on 28 June 2021, amended on 1 July 2022).

Clinical evaluations were performed in two sessions: T0 (24 h before BoNT-A administration) and T1 (10 days after injection). A time lapse of 10 days was chosen according to the delayed onset of the BoNT-A pharmacological effects, typically about 10 days [46]. To perform BoNT-A inoculation, an ultrasound guide (US) was used [24,44].

BoNT-A dose was systematically assessed by the Physical Medicine and Rehabilitation Specialist based upon the severity of spasticity measured with MAS, muscle volume, and the total number of muscles treated.

The post-injection treatment protocol did not differ from the usual clinical practice, consisting in an early performed 30-min session of electrical stimulation directed to the injected muscles. During the 10 days between injection and T1, subjects underwent the usual, patient-tailored, neuromotor rehabilitation program in a day-hospital regime.

Participation in the study did not cause adjunctive costs to the patients.

In order to evaluate potential systemic and sub-clinical side effects of BoNT-A, at T0 and T1, the contralateral upper limb strength was measured through a digital dynamometer (DynEx1, Akern s.r.l., Florence, Italy), and ANS parameters were collected throughout a 30-min ECG registration obtained with the VU University Ambulatory Monitoring System (VU-AMS, Vrije University, Amsterdam, Netherlands) [47].

Statistical analysis relied on the Wilcoxon–Mann–Whitney rank test for non-normal distributions.

The study diagram is reported in Figure 1.

The handgrip strength test was conceived to determine the maximum isometric strength produced by the hand and forearm muscles. In this study, it was performed on the healthy upper limb, contralateral to the one treated with BoNT-A, according to the Southampton protocol, as described by Roberts et al. [33].

To prevent possible confounders in ECG registration, patients were asked not to smoke, not to consume caffeine or alcohol, and to avoid heavy meals and physical activity for 2 h before each measurement. Moreover, the evaluations were carried out between 11.00 a.m. and 3.00 p.m. to minimize circadian rhythm alterations.

The ECG examination was performed in a quiet room with a steady temperature of 24 °C with VU-AMS machinery equipped with surface electrodes. The registrations were then analyzed through a specific software, “VU-AMS Data, Analysis and Management Software” (VU-DAMS program, Vrije University, Amsterdam, Netherlands), version 4.0, which evaluates HRV, ICG, and RSA. Each ECG recording was analyzed, and a 20-min frame was saved.

HRV is a valuation of heart rate fluctuations, based on the measure of the R-R interval between every two consecutive heart beats, called interbeat intervals (IBIs) [35,48]. It reflects the activity and the influence of the ANS on heart function. Thus, HRV can be considered as an index of neuro-cardiac function [49].

The ICG measures the electric thoracic bioimpedance detected by surface electrodes. Largely depending on fluid thoracic content, ICG reflects the variations of thoracic blood flow. Thus, ICG is a non-invasive, cheap, and safe method to evaluate hemodynamic parameters, such as stroke volume and the PEP [36]. The surface electrodes also detect electrocardiographic signals in order to synchronize the registration of the impedance with the heart activity, obtaining a detailed ANS analysis. We used a tetrapolar impedance measurement, the most broadly employed.

RSA represents the variation of heart rate in relation to respiration. It is a physiological phenomenon, according to which, heart rate increases during inspiration and decelerates in expiration, and it is an index of vagal activity [37].

## Figures and Tables

**Figure 1 toxins-14-00564-f001:**
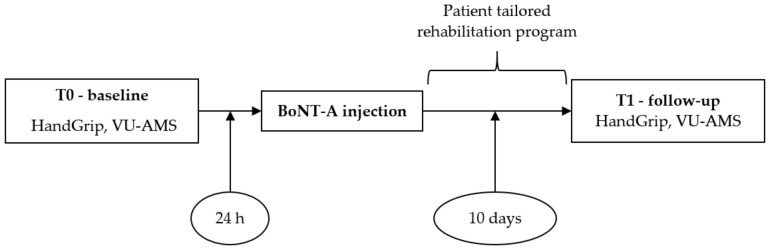
Study flowchart.

**Table 1 toxins-14-00564-t001:** Demographic characteristics and pharmacological treatment distribution.

	N = 15
Variable	N (%)
Sex	
Male (%)	9 (60%)
Female (%)	6 (40%)
Hemiparetic side	
Right	6 (40%)
Left	9 (60%)
Etiology	
Ischemic	11 (73.3%)
Hemorragic	4 (26.7%)
Mean age (SD)	60.3 years old (13.6)
Mean time from stroke (SD)	5.6 years (4.2)
Molecule	
IncobotulinumtoxinA (*n*)	2
AbobotulinumtoxinA (*n*)	2
OnabotulinumtoxinA (*n*)	11

**Table 2 toxins-14-00564-t002:** Botulinum Neurotoxin type-A distribution per patient (ID number 1–15), per injection site. LD: latissimus dorsi; SS: subscapularis; PM: pectoralis major; BB: biceps brachii; B: brachialis; BR: brachioradialis; PT: pronator teres; FCU: flexor carpi ulnaris; FCR: flexor carpi radialis; FDS: flexor digitorum superficialis; FDP: flexor digitorum profundus; FPL: flexor pollicis longus; FPB: flexor pollicis brevis; PL: palmaris longus; RF: rectus femoris; MG: medial gastrocnemius; LG: lateral gastrocnemius; SOL: soleus; TA: tibialis anterior; TP: tibialis posterior; FDL: flexor digitorum longus; FHL: flexor hallucis longus; EHL: extensor hallucis longus. All doses are in International Units (IU). I: IncobotulinumtoxinA; A: AbobotulinumtoxinA; O: OnabotulinumtoxinA.

		BoNT-A Dose per Patient, per Muscle														
	ID	1	2	3	4	5	6	7	8	9	10	11	12	13	14	15
Muscle																
*LD*		50														
*SS*											50					
*PM*		50				50			50	50			30		75	
*BB*		50	70				60		50	50			50		50	100
*B*		60	70				50		50	50			50		50	
*BR*		30	30				20								50	
*PT*					30	30	20		25				20		25	
*FCU*		50				30		25	25		25	25	20		50	100
*FCR*		50				30		25	25		25	25			50	100
*FDS*		25	50		50	30	70	25	25	20		30	25		50	150
*FDP*		25	30		50	30	70	25	25				25			150
*FPL*			20		50		30		25	20						100
*FPB*					20											
*PL*																100
*RF*		70	30													
*MG*		100	80	70		100	100		100	60		80	90	175		200
*LG*		100	70	60		100	80		100	50		60	70	150		100
*SOL*		100	80	70		100	100		100			80	90	175		200
*TA*			30			30										
*TP*			40			70	50									
*FDL*		40					50		50							100
*FHL*							50		50							100
*EHL*							50						30			
*Total dose (IU)*		800 O	600 O	200 O	200 I	600 O	800 O	100 I	700 O	300 O	100 O	300 O	500 O	500 A	400 O	1500 A

**Table 3 toxins-14-00564-t003:** Variations of the handgrip strength test and cardiorespiratory parameters from basal. No statistically significative modifications were found after BoNT-A treatment. SD: standard deviation; HRV: heart rate variability; ICG: impedance cardiography; IBIs: interbeat intervals; SDNN: standard deviation of normal-to-normal intervals; HR SD: heart rate standard deviation; RMSSD: root mean square of successive R-R intervals differences; LF: low frequency; HF: high frequency; LF/HF: ratio of LF-to-HF power; PEP: pre-ejection period; dz/dt min position: position of the minimum value of thoracic impedance; dz/dt min value: minimum value of thoracic impedance; RSA: respiratory sinus arrhythmia; RSA-0: the point where the RSA value is 0. Significance level: *p* < 0.05.

	T0 Mean (SD)	T1 Mean (SD)	*p*-Value
Handgrip [kg]	24.48 (10.47)	24.49 (9.93)	0.97
HRV			
IBIs number	1494.33 (267.90)	1580.07 (410.51)	0.15
IBIs average	831.47 (123.11)	805.70 (161.36)	0.23
SDNN	37.76 (15.49)	35.26 (19.86)	0.39
HR SD	3.20 (0.97)	3.41 (1.54)	0.45
RMSSD	23.16 (24.13)	19.78 (22.04)	0.25
LF	317.02 (345.35)	300.80 (352.07)	0.19
HF	86.58 (60.38)	173.20 (416.24)	0.36
LF/HF	4.50 (3.63)	4.02 (3.22)	0.72
ICG			
PEP	103.07 (24.30)	103.87 (23.97)	0.99
dz/dt min position	120.67 (25.55)	120.40 (28.88)	0.66
dz/dt min value	−0.66 (0.27)	−0.49 (0.33)	0.76
Respiration			
RSA	28.00 (13.46)	29.51 (20.22)	0.49
RSA-0	22.71 (9.89)	23.68 (17.08)	0.50

**Table 4 toxins-14-00564-t004:** Comparison between high-dose and low-dose subgroups, from T0 to T1, in every considered parameter. SD: standard deviation; HRV: heart rate variability; ICG: impedance cardiography; IBIs: interbeat intervals; SDNN: standard deviation of normal-to-normal intervals; HR SD: heart rate standard deviation; RMSSD: root mean square of successive R-R intervals differences; LF: low frequency; HF: high frequency; LF/HF: ratio of LF-to-HF power; PEP: pre-ejection period; dz/dt min position: position of the minimum value of thoracic impedance; dz/dt min value: minimum value of thoracic impedance; RSA: respiratory sinus arrhythmia; RSA-0: the point where the RSA value is 0. Since the distribution was not normal, a Mann–Whitney test was performed. Significance level: *p* < 0.05.

	N 4	N 11	
	High-Dose Group T1–T0 Mean Difference (SD)	Low-Dose Group T1–T0 Mean Difference (SD)	*p*-Value
Handgrip [kg]	−0.02 (0.14)	0.07 (0.13)	0.94
HRV			
IBIs number	0.04 (0.08)	0.05 (0.14)	0.73
IBIs average	−0.03 (0.08)	−0.01 (0.11)	0.62
SDNN	−0.29 (0.36)	0.06 (0.42)	0.94
HR SD	−0.15 (0.38)	0.15 (0.38)	0.98
RMSSD	−0.41 (0.37)	0.29 (1.26)	0.83
LF	−0.37 (0.52)	0.30 (1.40)	0.90
HF	−0.41 (0.37)	1,70 (5.23)	0.69
LF/HF	−0.12 (0.36)	0.06 (0.58)	0.76
ICG			
PEP	−0.05 (0.13)	0.12 (0.50)	0.63
dz/dt min position	−0.05 (0.24)	0.04 (0.29)	0.98
dz/dt min value	−0.21 (0.38)	−0.23 (0.44)	0.73
Respiration			
RSA	−0.05 (0.29)	0.19 (0.90)	0.83
RSA-0	−0.04 (0.32)	0.20 (0.97)	0.83

## Data Availability

The data presented in this study are available on request from the corresponding authors. The data are not publicly available due to the privacy protection policy.

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
