# Peer review of "The Lack of Systemic and Subclinical Side Effects of Botulinum Neurotoxin Type-A in Patients Affected by Post-Stroke Spasticity: A Longitudinal Cohort Study"

_toxins, 2022, doi:10.3390/toxins14080564_

Round 1

Reviewer 1 Report

The title use “Systemic diffusion and subclinical side effects of Botulinum Neurotoxin type-A….”, however, the results did not show the systemic diffusion and are consistent with the safety profile of use botulinum neurotoxin type A in patients with post-stroke spasticity. The authors claimed that “their study aims to investigate an additional population with a larger sample size” (lines 86-87), however, their study did not use an additional population, and the sample size is quite small, which did not match to their own study aim. Most of their discussions are based on the literature rather than their own results, and indeed,  their results did not provide any new information.

Reviewer 2 Report

Line 31. The extra space “and it  represents the fifth”

Lines 118-119. “Significance level: p <0.05. “ Maybe move "p<0.05" to the next line in its entirety?

Line245. “In our study sample we did not observe any subclinical sign of systemic BoNT-A diffusion.” And title of the article “Systemic diffusion…” When I read the title of the article, the first two words clearly caused me to expect data on pharmacokinetics. But I did not see any such data in the article. Indeed, there may be systemic diffusion of botulinum toxin, but it may not cause any visible effects due to lack of dose, for example. The doses studied in the article are small, and in my experience, I can say that it is difficult to expect systemic effects in doses less than 200 units. In animal experiments and measurement of botulinum toxin in the blood with a limit of quantitative detection of 0.15 units per milliliter of blood when injected intramuscularly up to 50 units into the rat femur, only some animals were found to have botulinum toxin in their blood. Perhaps the title of the article should be corrected somewhat? For example, "Systemic side effects and..." or "No systemic side effects and...".

Reviewer 3 Report

The authors followed the autonomic function of patients treated with BoNT-A with different total doses (100 U - 800 U onaBoNT-A, 500 and 1500 abo-BoNT-A, 100-200 incoBoNT-A), and examined contralateral grip strenght and measures of autonomic cardiorespiratory function. They found no differences before and after BoNT-A injections. 

Q1 - The toxin's effect on autonomic function may last for over a year and be considerably longer than the effects on hyperactive muscles. If the patients were treated with BoNT-A before the study, there could be residual effects from former injections and any possible change upon new treatment could be masked. In the M&M there is no data on the possible treatment status of patients before the T0. Were the patients treatment-naive before the study i.e. did they receive BoNT-A injections before? 

Q2 - the dosages across the patients vary significantly. It is more likely that some effects will be seen in patients treated with high - above the FDA-recommended maximal dose patients. Can you please give percent change (T1/T0) for each parameter in individual patients treated with high ona (800, 700) and aboBoNT-A dose (1500) and compare it with patients treated with low doses?

Round 2

Reviewer 1 Report

The authors' response is sufficient, though the findings in this manuscript are only supportive to the current literatures.